# Staphylococcal Enterotoxins Enhance Biofilm Formation by *Staphylococcus aureus* in Platelet Concentrates

**DOI:** 10.3390/microorganisms11010089

**Published:** 2022-12-29

**Authors:** Sylvia Ighem Chi, Sandra Ramirez-Arcos

**Affiliations:** 1Medical Affairs and Innovation, Canadian Blood Services, Ottawa, ON K2E 8A6, Canada; 2Department of Biochemistry, Microbiology and Immunology, University of Ottawa, Ottawa, ON K1N 6N5, Canada

**Keywords:** biofilm formation, platelet concentrates, staphylococcal enterotoxins

## Abstract

Biofilm formation and slow growth by *Staphylococcus aureus* in platelet concentrates (PCs) cause missed detection of this bacterium during routine PC screening with automated culture systems. This heightens the chances of false-negative screening transfusions and pre-disposes transfusion patients to an elevated risk of sepsis due to secretion of staphylococcal enterotoxins (SEs) in PCs. A hybrid approach of comparative RNAseq analyses and CRISPR mutagenesis of SE genes was employed to investigate the effect of SEs in *S. aureus* growth and biofilm formation in PCs. RNAseq data showed no differential expression for key biofilm genes, whereas SE genes were upregulated (>0.5- to 3.6-fold change) in PCs compared to trypticase soy broth (TSB). Remarkably, growth and biofilm formation assays revealed increased growth for the *S. aureus* SE mutants, while their ability to form biofilms was significantly impaired (−6.8- to −2.4-fold change) in comparison to the wild type strain, in both PCs and TSB. Through the well-established superantigen mechanism of SEs, we propose three roles for SEs during biofilm development in PCs: (1) provide a scaffold for biofilm matrix, (2) mediate cell-to-cell aggregation, and (3) guarantee biofilm survival. Furthermore, SE contribution to both growth and biofilm development seems to be centrally regulated by *agr* via quorum sensing and by *saeSR* and *sigB*. This study reveals new roles for SEs, which enforce their relevance in ensuring PC safety for transfusion patients. It further deciphers the underlying reasons for failed *S. aureus* detection in PCs during screening with automated culture systems.

## 1. Introduction

Bacterial biofilms pose a serious challenge to healthcare, medical, and food industries globally. Biofilms are complex surface-attached aggregates of cells encased by a matrix made of polysaccharides, proteins, and extracellular DNA (eDNA) [1]. Bacteria can establish biofilms on biotic surfaces like epithelial cells, as well as abiotic surfaces of medical devices [2,3] as a means to shield from antimicrobials and immune clearance [4]. Several Gram-positive and Gram-negative bacteria are capable of producing biofilms in diverse environments and settings, with the Gram-positive virulent pathogen *Staphylococcus aureus* being a predominant threat to healthcare, including transfusion medicine. 

*S. aureus* produces several virulent factors, including exotoxins and biofilm-associated proteins encoded by genes that are controlled by a network of global regulators (*agr*, *mgrA*, *saeSR*, *sarA*, *sigB*, etc.) that mediate resilience and persistence of this organism in human infections [5,6]. Biofilms are produced by 99.2% of *S. aureus* isolates and are formed through a complex process comprising five main steps [7,8]. At stage I, *S. aureus* reversibly attach to biotic surfaces via microbial surface components recognizing adhesive matrix molecules (MSCRAMMs) or hydrophobically to abiotic surfaces, setting the scaffold for stage II, which is marked by adherence of bacterial cells to the surfaces using bacterial adhesins and eDNA. Then, nuclease degradation and release of eDNA allow a massive exodus of cells while the remaining cells aggregate in three-dimension microcolonies (stage III). At stage IV, cells within the microcolonies rapidly divide, maturing the structure and stabilizing into robust aggregates of proteins, phenol soluble modulins (PSM), and eDNA. Finally, at stage V, when confluence is attained, the biofilm matrix is modulated by proteases and/or PSM mediated by quorum sensing (QS) *agr*, thereby freeing bacterial cells to an independent free-floating (planktonic) lifestyle. *S. aureus* utilizes four known mechanisms to form biofilms: (i) biofilm formation dependent on expression of the *icaADBC* operon that encodes for enzymes associated with synthesis of the polysaccharide intracellular adhesin (PIA). IcaAD has glycosyltransferase activity, whereas IcaC appears to be involved in externalization of the growing polysaccharide. The IcaB sequence shares homology with deacetylases, indicating that this protein may be involved in polysaccharide modification; (ii) *ica*-independent biofilm formation with a matrix formed mainly by eDNA and surface proteins such as BAP, FnBPs, and SasG/Aap; (iii) coagulase-mediated biofilm formation that involves conversion of fibrinogen into fibrin for biofilm scaffold; and (iv) PSM-dependent biofilm formation, where the surfactant properties of PSM enhance biofilm accumulation by forming amyloid aggregates, and help with biofilm dispersals [9,10].

In transfusion settings, platelet concentrates (PCs) are a therapeutic product used to treat patients with platelet dysfunction and bleeding disorders. In PCs, proliferation of bacteria introduced during blood collection is enhanced due to the PC storage conditions that are required to maintain platelet functionality, and include incubation at 20 ± 2 °C, under agitation, in nutrient-rich solutions with a neutral pH, which are stored in oxygen permeable containers for up to seven days [11]. Several strategies have been implemented to mitigate the risk of transfusing bacterially contaminated PCs, including a blood donor questionnaire to determine if the donor has had a recent infection; blood donor skin disinfection with chlorhexidine and isopropyl alcohol or an alternative method; diversion of the first 30 to 40 mL of collected blood (i.e., first aliquot diversion), where it has been demonstrated that a skin plug with skin flora bacteria resides; and screening of PCs with culture methods or rapid assays, or treatment of PCs with pathogen reduction technologies [11].

Despite the implementation of a stringent donor skin disinfection and first aliquot diversion to mitigate the risk of transfusing bacterially contaminated PCs, bacteria can still be present in this blood component and escape screening with culture methods. This is mostly due to fact that disinfection cannot reach the deeper layers of the skin where some bacteria reside, forming biofilms [11]. The predominant bacteria isolated from contaminated PC units are skin flora organisms, such as propionibacteria and staphylococci. Importantly, among the frequently isolated bacterial contaminants from PCs, *S. aureus* is remarkable for causing transfusion-associated complications and sepsis, as this pathogen can form biofilms during PC storage [12,13]. Furthermore, delayed growth and biofilm production by *S. aureus* in PCs are the chief causes of missed detection of this bacterium during PC screening with automated culture systems [11,12,13]. This heightens the chances of false-negative screening transfusions and pre-disposes PC transfusion recipients to an elevated risk of sepsis [12,13], which is exacerbated by the secretion of exotoxins in the PC milieu [12,13].

Within the exotoxins, staphylococcal superantigens (SAgs) are noted as predominant virulence factors of *S. aureus* [14] secreted by approximately 80% of methicillin-susceptible *S. aureus* and >90% of methicillin-resistant *S. aureus* isolates [15]. SAgs are categorised into two broad classes: superantigen-like toxins (SSLs), which inhibit host immune responses, and staphylococcal enterotoxins (SEs), responsible for septic shock symptoms [15]. SEs are a group of 26 heat- and acid-resistant proteins, including SEG and SEH, which are secreted mostly at the post-exponential growth phase of *S. aureus*, and have the ability to induce emesis and superantigenicity [15,16]. In contrast to conventional antigens, SAgs interfere with immune functions that combat bacterial infection by binding non-specifically to major histocompatibility complex class II molecules on antigen presenting cells and T-cell receptors, which stimulate a vast number of T-cells causing cytokine release [6].

Besides emetic and superantigenic activities, recent discoveries unveiled additional roles for SEs, including involvement in chronic rhinosinusitis and nasal polyposis [17]; these exotoxins have also been shown to have anti-cancer properties [18]. A study by Calabrese and colleagues reported SEs associated with asthma as specific IgE to staphylococcal enterotoxins (_SE_-IgE) was frequently detected in late onset of severe asthma patients [19]. These emergent findings highlight the potential for SEs to play other unknown functions. Almost all *S. aureus* isolates produce biofilms, and over 80% encode SEs, yet knowledge about the relationship between SE production and biofilm development is limited. 

In this study, we investigated the novel relationship between SEs and growth, and SEs and biofilm formation by *S. aureus* in PCs using transcriptomics and molecular biology approaches. As we have previously shown that the PC storage environment triggers biofilm formation by *S. aureus* [12,13], we hypothesized that SE production is enhanced during PC storage and is directly linked to increased biofilm formation. The main objective of the study was to compare expression of *S. aureus* SE and biofilm-associated genes in PC cultures against bacteria grown in trypticase soy broth (TSB). The secondary objective was to analyze the differential gene expression of regulators and propose a model for the modulation of *S. aureus* growth and biofilm development in PCs.

## 2. Materials and Methods

The overall experimental design of this study is described in Figure 1.

### 2.1. Platelet Concentrates

The PC units used in this study were manufactured in 100% plasma at the Blood4Research Facility of Canadian Blood Services (netCAD, Vancouver, BC, Canada) in agreement with standard procedures. The PC units were shipped to the Canadian Blood Services Microbiology laboratory in Ottawa, Canada. The Canadian Blood Services Research Ethical Board granted ethical approval for this study.

### 2.2. Bacterial Strains, Plasmids, and Growth Conditions

The *S. aureus* strains used in this study were isolated from contaminated PCs in Canada and England as recently described [20,21,22,23]. Two of the isolates, CBS2016-05 and PS/BAC/169/17/W, are strong biofilm producers, while the other two strains, CI/BAC/25/13/W and PS/BAC/317/16/W, form weak biofilms [24]. *E. coli* DC10B (Thermo Scientific, Waltham, MA, USA) and a CRISPR-based *E. coli*/*S. aureus* temperature sensitive high copy number plasmid (pCasSA), designed for genome editing in *S. aureus* (Addgene, Watertown, MA, USA), were used to generate SE mutants in *S. aureus* CBS2016-05. *S. aureus* strains were cultured in TSB at 37 °C with agitation for growth assays, and in TSB supplemented with 0.5% glucose (TSBg) under static conditions for biofilm formation analyses. In PCs, bacteria were incubated under PC storage conditions (20 ± 2 °C, under agitation in a PC incubator). *E. coli* DC10B and *S. aureus* CBS2016-05 carrying pCasSA or derived plasmids were grown at 30 °C.

### 2.3. RNA Isolation, Next Generation Sequencing, and Data Analyses

The RNAseq dataset of the four strains grown separately in TSB and in PCs were next generation sequenced from total RNA submitted to the StemCore Laboratories Ottawa hospital (http://www.ohri.ca/stemcore/, accessed on 24 December 2022). The total RNA was extracted from *S. aureus* grown to stationary phase in PCs and in TSB with initial inocula of ~10^6^ colony forming units (CFU)/mL under PC storage conditions (20 ± 2 °C, under agitation in a PC incubator). The bacterial cell pellets, obtained by centrifuging the cultures, were homogenized using a FastPrep^®^ Instrument before proceeding with RNA extraction with a Fast RNA Blue kit (MP Biomedicals, Solon, OH, USA), per the manufacturer’s instructions. Phase separation was achieved with chloroform, followed by precipitation of RNA with ethanol at −20 °C for a minimum of 1 h, and then the pellet was washed twice in 75% ethanol prior to rehydration. Subsequently, RNA obtained from the PC samples was depleted of mammalian RNA using a MICROBEnrich^TM^ kit (Invitrogen, Waltham, MA, USA). RNA obtained from TSB and PC samples was DNase-treated with a TURBO DNase kit (Invitrogen, Waltham, MA, USA). RNA samples with RNA integrity number (RIN) ≥ 8 were depleted of rRNA, and complementary DNA (cDNA) was synthesized. Then, 300 base pair cDNA illumina libraries were prepared using a TrueSeq stranded mRNA preparation kit procedure (Illumina, San Diego, CA, USA). Finally, both samples obtained from TSB and PCs were separately sequenced in triplicate on a NextSeq 500 sequencing system (Illumina, San Diego, CA, USA). Core differential expression (DE) analyses were performed between the PCs and TSB samples for each strain via DESeq2 pipeline at the Core Facilities at University of Ottawa and the Ottawa Hospital Research Institute (https://www.ohri.ca/bioinformatics/ (accessed on 24 December 2022)). The transcriptome dataset for DE biofilm genes and other related virulence determinants was further analyzed with a cutoff at ±0.5-fold change. RNAseq data for the four *S. aureus* strains and three replicates (a, b, and c) prepared in PCs and TSB were submitted to the National Center for Biotechnology Information of the National Library of Medicine (BioProject number PRJNA915492). Information about accession numbers and release dates is available at https://dataview.ncbi.nlm.nih.gov/object/PRJNA915492?reviewer=413fkhgmqdmj42p7u0ursh41tt (accessed on 24 December 2022).

### 2.4. Quantitative Reverse Transcription PCR Verification

For quantitative reverse transcription PCR (qRT-PCR), approximately 1 ug of total RNA was subjected to reverse transcription (RT) with the QuantiTect Reverse Transcription kit protocol (Qiagen, Germantown, MD, USA) after treatment with TURBO DNAse (Invitrogen, Waltham, MA, USA). The RT–qPCR reaction mixture, composed of the purified RNA, RT buffer, RT primer mix, and Quantitect reverse transcriptase, was incubated at 42 °C for 30 min. A total of 2 uL (10 ng) of the cDNA was subsequently added to the qPCR mixture comprising 5 uL of 2X SYBR Green PCR master mix from QuantiNova SYBR Green PCR kit (Qiagen, Germantown, MD, USA), 1 μL of forward and reverse primers (0.7 μM), and 2 uL nuclease-free water. Each gene and condition were run in duplicate using equal amounts of cDNAs in TSB and PCs. For controls, *gyrA* (positive) and nuclease-free H_2_O (negative) were used. A CFX96 Touch^TM^ Real-Time PCR Detection System (BioRad, Hercules, California, CA, USA) was used to run the experiments. Transcript copy numbers were determined using concentrations generated by mean Cq (quantification cycle) values of the samples obtained by standard curve analyses based on 10-fold dilutions of their respective PCR amplicons of known concentrations.

### 2.5. Generation of S. aureus Exotoxin Mutants

CRISPR-Cas9 mutagenesis technique using a CRISPR-based *E. coli*/*S. aureus* temperature sensitive high copy number plasmid, pCasSA (Addgene, Watertown, MA, USA), was exploited for generating *seg* and *seh* knockout mutants (Δ*seg* and Δ*seh*) following a previously described procedure [25], with some modifications. Briefly, homologous region (HR) fragments with overlapping 30 to 40 bp from upstream and downstream regions of *seg* and *seh* were generated by PCR amplification (using OneTaq DNA polymerase (NEB, Ipswich, MA, USA) with 30 bp overhangs motifs specific for pCasSA reverse amplification. The PCR-based HR amplicons were cloned into linearized pCasSA using a Gibson assembly Master mix kit (NEB, Ipswich, MA, USA), and the constructed recombinant plasmids (pCasSA+HR) carrying the inserts were heat-shocked transformed into *E. coli* DC10B. Positive clones were selected on Luria broth agar (LBA) plates supplemented with 50 mg/mL of kanamycin after incubation at 30 °C for 36 h and confirmed by colony PCR using a OneTaq DNA polymerase kit (NEB, Ipswich, MA, USA). pCasSA+HR DNA was extracted from the confirmed clones with a QIAprep Spin miniprep kit (Qiagen, Germantown, MD, USA). Subsequently, small guide RNA (sgRNA) spacers were cloned into the pCasSA+HR plasmids with a golden gate assembly kit (NEB, Ipswich, MA, USA). The resultant plasmids (pCasSA+HR+sgRNA) were transformed into *E. coli* DC10B, and positive clones were PCR screened, followed by pDNA isolation as described above. The purified pCasSA+HR+sgRNA plasmids were electroporated into *S. aureus* following a reported protocol [26]. *S. aureus* SE mutants (Δ*seg* and Δ*seh*) were screened on trypticase soy agar (TSA) plates supplemented with 10 mg/mL chloramphenicol, verified by colony PCR, and subjected to plasmid curation by incubation at 42 °C for 3 h. Sanger sequencing further confirmed the deletion of the *seg* and *seh* genes in *S. aureus* CBS2016-05. Polar effects of the *seg* mutation on the upstream *sen* gene and downstream *hypo*, *pepA1*, *IS3* and *hypo* genes were investigated by PCR. Similarly, PCR was used to investigate polar effects of the *seh* mutation on the upstream *exo* gene and downstream *IS3* and *trans* genes.

### 2.6. Growth Dynamics of Wild Type S. aureus and Mutant Strains

Growth dynamics of wild type *S. aureus* CBS2016-05 and Δ*seg* and Δ*seh* mutants were assessed in TSB and in PCs. Initial inocula (OD_600_ = 0.002), which corresponds to approximately 10^6^ CFU/mL, were used for TSB and PC cultures. The TSB cultures were incubated at 37 °C with shaking at 160 rpm and growth was measured every 2 h from time point zero to 14 h of incubation based on optical density OD_600_ on a Ultrospec 3100 pro UV/Visible spectrophotometer (Biocompare, South San Francisco, CA, USA). Samples were taken at the same time points for 10-fold serial dilutions prepared in TSB, plated on TSA in duplicate, and then incubated overnight at 37 °C for determination of the bacteria concentration (CFU/mL). Samples from spiked PC cultures, which were grown under PC storage conditions (20 ± 2 °C, under agitation in a PC incubator), were taken every 24 h for serial dilutions prepared in TSB, plated on TSA in duplicate, and then incubated overnight at 37 °C for determination of the bacteria concentration (CFU/mL). The experiments were performed in triplicate with three biological replicates.

### 2.7. In Vitro Biofilm Assessments of Wild Type S. aureus and Mutant Strains

Biofilm formation of wild type *S. aureus* CBS2016-05 and Δ*seg* and Δ*seh* mutants was investigated by performing semi-quantitative crystal violet biofilm assays as per established procedures [27] with minor changes. *S. aureus* ATCC 25923 was used as positive control for biofilm formation. In brief, TSBg and PCs were inoculated with wild type and mutant strains and aliquoted into 6-well plates at an initial inocula of approximately 10^8^ CFU/mL (OD_600_ = 0.1). Following 24 h of static growth at 37 °C in TSBg and 120 h of growth in PCs under PC storage conditions (20 ± 2 °C, under agitation in a PC incubator), biofilms were washed three times in 4 mL PBS and stained for 30 mins with 3 mL crystal violet (Lifesupply, Britsih Columbia, CA, USA). The plates were washed as above, followed by an addition of 3 mL of glacial acetic acid (Lifesupply, Britsih Columbia, CA, USA) to solubilize biofilms. The absorbance of the elutes was measured at OD_492_ with a spectra Max 190 microtiter plate reader 190 equipment (Molecular Devices, San Jose, CA, USA). The baseline readings of TSBg and PCs with no bacteria were subtracted from the readings at OD_492_. Crystal violet assays were performed in biological triplicates with three technical replicates.

### 2.8. Statistical Analyses

Two-tailed *t* tests were used to compared biofilm gene expression, growth dynamics, and biofilm formation between the wild type *S. aureus* CBS2016-05 and Δ*seg* and Δ*seh* mutant strains. Growth curves were additionally analysed with the online tool “Compare Groups of Growth Curves” (Compare Groups of Growth Curves (wehi.edu.au (accessed on 24 December 2022))).

## 3. Results

### 3.1. Virulence Factors Involved in Biofilm Formation Are Differentially Expressed in PCs

Data obtained from RNAseq of four *S. aureus* strains cultured in PCs and in TSB showed that the expression of biofilm-associated genes (Table 1) and other virulence factors (Table 2) varied between strains. While some factors, like surface adhesins, were upregulated in some isolates, in others, they were downregulated (Table 2). However, expression of most exotoxins and capsule was consistently upregulated in all strains (Table 2). The positive regulators of enterotoxins, *sigB*, *saeSR*, and *agr*, that also control stress, nutrient availability, and cell density, were downregulated, while biofilm enhancers *sarA* and *rot*, which repress SE expression, were upregulated in PCs compared to TSB (Table 2).

Surprisingly, there were some discrepancies with the expression of *ica*-biofilm genes within the four *S. aureus* strains. The *ica* genes of strong biofilm producer *S. aureus* strains appeared repressed in PCs, whereas they were upregulated in the weak biofilm strains (Table 1 and Figure 2). Importantly, Quantitative RT-PCR analyses revealed no significant differences in the expression of the biofilm genes between the wild type and mutant strains (Figure 3). A two-tailed *t* test confirmed that there were no significant differences in the expression of the biofilm genes between the wild and mutant strains in both PCs and TSB (*p* > 0.05).

### 3.2. S. aureus Biofilm Establishment Is Regulated Based on Niche and Strain Background

Following RNAseq revelation of *ica*-operon repression in wild type *S. aureus* strains that are strong biofilm formers, we then resolved to investigate genes associated with other biofilm production mechanisms from our RNAseq data set of all four *S. aureus* isolates. Our analyses showed that biofilm production is strain-specific and likely involves a complex interplay of at least two mechanisms. None of the four isolates seemed to use the amyloid pathway, as the associated genes (PSM) were significantly downregulated in PCs, while protein/eDNA and fibrin pathways were upregulated in all strains (Figure 2). It is apparent that *S. aureus* CI/BAC/25/13/W and PS/BAC/317/16/W are *ica*-dependent as *ica* gene expression showed a significant increase in these strains (Figure 2B,D, Table 1). The remaining two isolates (CBS2016-05 and PS/BAC/169/17/W) had higher expression of *sasC*, *cidA*, and *lrgB*, as well as *clfA*, *clfB*, *coa*, *fnbB*, *sdrC*, and *sdrD* (1 ≤ 2 log2-fold change), which are connected to protein/eDNA and fibrin-mediated mechanisms, respectively (Figure 2A,C, Table 1).

### 3.3. Staphylococcal Enterotoxins Impaired Growth of S. aureus in PCs

Based on significant expression differences of SEs between PCs and TSB, Δ*seg* and Δ*seh* mutants were created with CBS2016-05 as a background strain, and their effect on growth dynamics of *S. aureus* was assessed in comparison to the wild type in TSB and in PCs. Our results show that all bacteria reached higher concentrations in PCs (approximately 10^14^ CFU/mL) compared to TSB (approximately 10^10^ CFU/mL). Importantly, mutant strains grew faster than the wild type strain in both PC and TSB growth environments (Figure 4). This was observed since the beginning of the growth curves, at 2 h in the TSB cultures, and 24 h in the PC cultures. A difference of approximately 2 Log between wild type and mutant strains was maintained throughout the growth curves; however, this difference was not statistically significant as demonstrated by a two-tailed *t* test analysis (*p* > 0.05). There was no difference in growth between the *S. aureus* Δ*seg* and Δ*seh* mutants.

### 3.4. Staphylococcal Enterotoxins Significantly Enhanced S. aureus Biofilm Formation in PCs

The potential impact of *S. aureus* SE on biofilm formation was assessed. Our data showed a significant reduction of biofilm formation in the Δ*seg* and Δ*seh* strains in comparison to wild type *S. aureus* CBS2016-05 (2.4- to 6.8-fold reduction) in both PCs and TSBg (*p* < 0.05) (Figure 5). While wild type *S. aureus* CBS2016-05 produced strong biofilms with readings at OD_492_ of approximately 2.0 in TSBg, the mutant strains yielded absorbances of approximately 0.5 at OD_492_ in TSBg (Figure 5). As demonstrated previously by our team, *S. aureus* produces stronger biofilms in PCs compared to TSBg cultures [12,13]. Wild type *S. aureus* CBS2016-05 produced biofilms with readings of up to 9.0 at OD_492_ in PCs, while the mutant strains had reduced readings at OD_492_, approximately 1.0 and 3.0 for the Δ*seh* and Δ*seg* strains, respectively (Figure 5). PCR analyses revealed no polar effects of the *seg* and *seh* mutations on upstream and downstream genes (data not shown).

## 4. Discussion

Delayed growth and biofilm production are eminent escape strategies employed by *S. aureus* to avert detection in PCs during screening with automated culture systems [11], which heightens the chances of false-negative transfusions events. We have shown that patients transfused with PCs contaminated with *S. aureus* have an elevated risk of sepsis due to the presence of SEs in this blood product [12,13]. It is usually assumed that septic transfusion reactions characterized by a rapid onset of symptoms such as fever, chills, and hypotension are due to the presence of Gram-negative bacteria in blood components, such as PCs. This is due to the inflammatory response triggered by the presence of endotoxins. However, we have shown that transfusion of PCs contaminated with Gram positive bacteria, such as *S. aureus*, can also trigger typical septic transfusion symptoms due to the release of exotoxins during PC storage. PCs contaminated with exotoxin-producing *S. aureus* were responsible for a septic transfusion case involving a unit with obvious visual changes characterized by a fibrous clot, which contained platelet debris and clusters of bacterial cells [12]. More recently, we reported another septic transfusion case implicating a PC unit contaminated with an exotoxin SEG-producing, biofilm-forming *S. aureus* strain [13]. In both cases, the PC units had been screened with culture methods that missed detection of the contaminant bacteria, highlighting the risk posed by this species to susceptible transfusion patients, who are usually immunocompromised. The present study advances knowledge on the dynamics of growth and biofilm formation of *S. aureus* in PCs. We provide novel information regarding an interaction thus far not demonstrated between the effect of SE production during PC storage and the modulation of growth and biofilm formation. Derivative SE mutants showed acceleration in growth, while biofilm development was significantly reduced compared to the wild type strain. 

Staphylococcal exotoxins, alpha toxin (*hla*), lukocidin AB (*lukAB*), and staphylococcal protein A (*spa*) have been reported to enhance biofilm establishment in vivo via superantigenicity and mediate cell-to-cell adhesion [28,29]. In endocarditis, *hlb* promotes biofilm matrix accumulation by forming covalent cross-links with itself in the presence of DNA, as well as through its sphingomyelinase activity [30,31]. It is possible that SEs contribute to biofilms using a similar mechanism not yet described in the specific PC milieu. A superantigen-like protein was shown to bind platelet glycoproteins [32]. Moreover, platelets express T-cell co-stimulatory molecules [33] and, in an infection state, they have been reported to produce MHCclass II molecules [34], which are both plausible substrates for SEs superantigenic binding. Through SE superantigenic activity, we propose three roles that enterotoxins might play in the developmental steps of biofilm establishment by *S. aureus* in PCs (Figure 6). (1) *Superantigenic complex provides a scaffold for matrix framework* via unconventional binding of SEs to yet unknown molecules that could be expressed by activated platelets during PC storage simulating scenarios seen during infections with *S. aureus*. (2) *SEs mediate cell-to-cell aggregation* by activating platelet cells, which become sticky and adhere to each other, enhancing biofilm accumulation [12]. (3) *SEs ensure survival of S. aureus biofilms* by inducing cytokine release that inhibits the functions of immune defenders [35].

Is SE modulation of bacterial growth connected to biofilm production?

At the cellular level, *S. aureus* has two options: either to increase proliferation and be exposed to host immune clearance or slow it down and enhance survival mechanisms. The choice made seems to depend on the host environment and availability of nutrients. While growth involves energy expense and may be favored in the natural commensal nutrient-rich habitat, survival is critical and preferred in harsh host environments like the one provided by PCs. Bacterial cell proliferation is influenced by environmental factors, such as nutrients and stress. Depending on the conditions, *S. aureus* utilizes a network of virulence regulators like *sigB* (alternative sigma factor B, expresses in stressful environments), a nutrient regulator, *saeSR* (*S. aureus* exoprotein expression), *sarA* (staphylococcal accessory regulator), a promoter of biofilms, and *agr* (accessory gene regulator) that controls cell density via QS [6]. These global regulators seemed to control the expression of biofilm and SE genes in several ways. The main positive regulators of enterotoxins, *agr* and its effector RNAIII (*hld*), were downregulated while *rot*, a repressor of toxins and positive regulators of biofilm genes, had slightly increased expression in the wild type in PCs than in TSB, according to RNAseq analyses.

Based on our findings, the observed differences in growth and biofilm accumulation between the wild type and derivative toxin mutants could be explained by an interplay of regulatory networks centrally controlled by three relevant regulators: *saeSR*, *sigB*, and QS *agr*, that indirectly control the expression of both SE and biofilm genes. Here, we propose two scenarios that link *S. aureus* growth dynamics to its biofilm production in connection to SEs (Figure 7): (I) in high cell density and nutrient deficient environment, *agr* is alarmed via QS to signal expression of SEs by repressing *rot* through expression of its effector RNAIII. When SEs are expressed, their superantigenic activity enhance biofilm formation as proposed above. Moreover, the superantigenic activity triggers an outburst of cytokines making the cell toxic, which in turn impairs *S. aureus* growth. (II) When cell density is low in nutrient-rich conditions, *saeSR* is expressed, which in turn activates *sigB* to repress *agr* signalling, which consequentially decreases toxin production. In this state, the toxin scaffold for the biofilm structure is limited, platelets would not be activated, and cell toxicity becomes low, allowing unhindered host immune clearance that eventually destabilizes the accumulated biofilms. In addition, the bacterial cells continue to proliferate, consuming the available nutrients until the cell density increases and *agr* is activated again.

The results of this study are based on analyses of four transfusion relevant *S. aureus* isolates and could be complemented with the addition of other strains, including *S. aureus* isolated from clinical samples, to understand if the gene expression changes observed in our transcriptome analyses are unique to bacteria obtained from contaminated PCs. Furthermore, it would be interesting to investigate if other SEs play a similar role as the one observed for SEG and SEH in modulating growth and biofilm formation of *S. aureus* in PCs.

The data presented herein advance knowledge by providing a new modulatory role of SEs on growth and biofilm formation by *S. aureus.* Future studies should be focused on the application of this new knowledge to improve the safety of transfusion patients. Specifically, we propose to confirm our proposed model which depicts platelet activation triggered by SEs during PC storage with consequential cytokine release, and promotion of biofilm formation. We also suggest complementing current culture-based PC screening with SE detection, as interdicting PC units that contain SEs would prevent septic transfusion reactions.

## 5. Conclusions

The principal reasons for failed *S. aureus* detection during PC screening with culture methods are slow growth and biofilm formation, which are characteristic of this species in contaminated PC units. However, mechanisms of biofilm formation are strain-dependent, as demonstrated by our transcriptome analyses; while some strains produce PIA-based biofilms, others form fibrin-based or protein/eDNA-based biofilms. Through a mutagenesis approach, we provide novel findings of staphylococcal enterotoxins modulation of *S. aureus* growth and biofilm formation in the unique PC storage environment. Based on our transcriptome data, we propose that growth and biofilm formation in PCs are regulated by the accessory gene regulator (*agr*) quorum sensing system that is affected by cell density, and which positively regulates expression of enterotoxin and biofilm-related genes. Our observations support a model where enterotoxin production decreases the growth rate and promotes biofilm formation of *S. aureus* in PCs. By proposing this multifaceted role of SEs in *S. aureus* growth dynamics and biofilms, we demystify underlying reasons for *S. aureus* missed detection during PC screening with automated culture systems. This study reveals new roles of SEs and enforces their relevance in the clinical outcomes of transfusion patients receiving contaminated PC units with *S. aureus*. It is therefore important to consider complementing current PC screening methods with enterotoxin detection to improve PC safety.

## Figures and Tables

**Figure 1 microorganisms-11-00089-f001:**
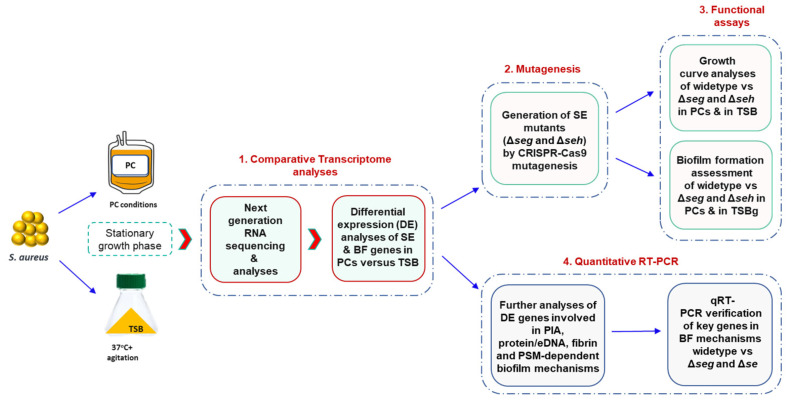
Methodologies used in this study. (1) RNA extracted from *S. aureus* cultures grown to stationary phase in PCs or TSB was subjected to next generation RNA sequencing. (2) Based on differential gene expression, genes encoding for enterotoxins SEG and SEH were deleted using CRISPR-Cas9 mutagenesis. (3) Functional assays of the wild-type and mutant strains, including growth curves and biofilm formation, were performed. (4) Further bioinformatic analysis and qRT-PCR analyses were conducted to propose scenarios for modulation of *S. aureus* growth and biofilm development in PCs. PC; platelet concentrate, TSB; trypticase soy broth, TSBg; trypticase soy broth supplemented with 0.5% glucose, SE; Staphylococcal enterotoxin, BF; biofilm formation, PIA; polysaccharide intracellular adhesin, and PSM; phenol soluble modulin.

**Figure 2 microorganisms-11-00089-f002:**
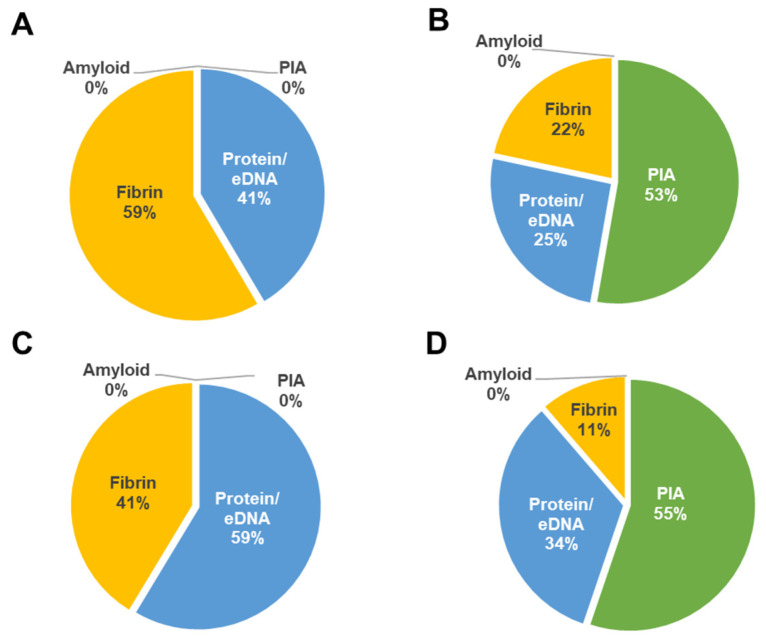
RNAseq-based percentage upregulation of the different mechanisms involved in biofilm formation in studied *S. aureus* isolates. (**A**) CBS2016-05, (**B**) CI/BAC/25/13/W, (**C**) PS/BAC/169/17/W, and (**D**) PS/BAC/317/16/W. RNAseq data was obtained from the strains grown in PCs and in TSB. The percentages were deduced from the calculated values of the number of genes with enhanced expression in PCs compared to TSB out of the sum of the known genes involved in each biofilm formation mechanism.

**Figure 3 microorganisms-11-00089-f003:**
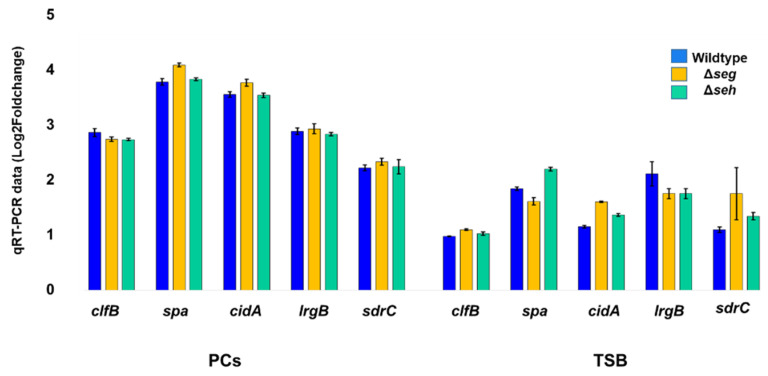
Graph representing quantitative reverse transcription PCR (qRT-PCT) results of candidate biofilm genes in *S. aureus* strain CBS2016-05 and derivative Δ*seg* and Δ*seh* deletion mutants. The complementary DNA samples for this assay were prepared from the S*. aureus* strains cultured in TSB and in PCs. TSB; trypticase soy broth (TSB), PC; platelet concentrates. (N = 2). Error bars are standard deviation based on two replicates.

**Figure 4 microorganisms-11-00089-f004:**
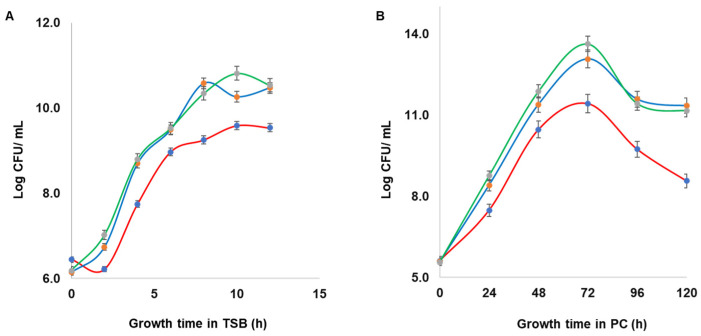
Growth curve analyses of *S. aureus* CBS2016-05 wild type and derivative enterotoxin mutant strains. The bacterial strains were grown in (**A**) trypticase soy broth (TSB) at 37 °C with agitation, and (**B**) platelet concentrates (PCs) incubated 20 ± 2 °C under agitation. N = 3; error bars represent standard deviation based on three replicates.

**Figure 5 microorganisms-11-00089-f005:**
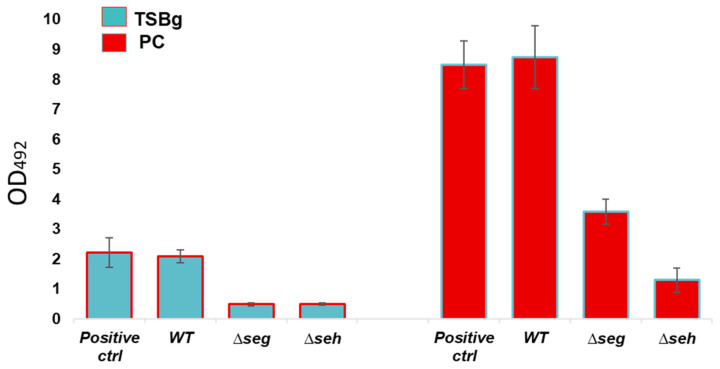
Biofilm formation assessment of *S. aureus* CBS2016-05 wild type versus derivative enterotoxin mutant strains. The bacterial strains were incubated in trypticase soy broth supplemented with 0.5% glucose (TSBg) at 37 °C in static conditions for 24 h and in platelet concentrates (PCs) incubated at 20 ± 2 °C under agitation for 5 days. N = 3; error bars are standard deviation based on three replicates. *S. aureus* ATCC 25923 (positive control strain).

**Figure 6 microorganisms-11-00089-f006:**
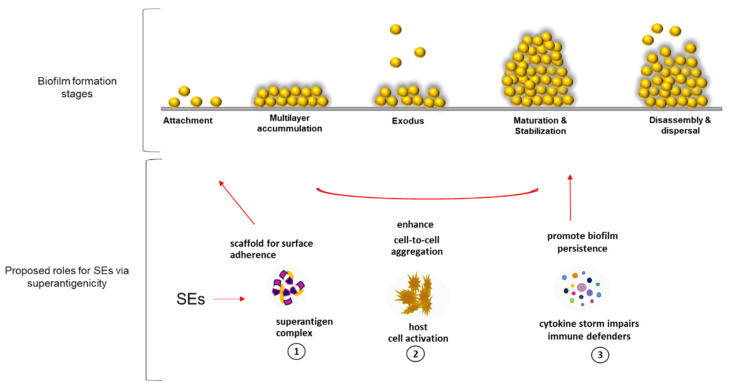
Proposed roles played by SEs via superantigenic activity in the developmental stages of *S. aureus* biofilm. (1) Superantigen complex formation provides a scaffold for matrix framework, (2) mediate cell-to-cell aggregation by activating platelet cells, which become sticky and adhere to each other, enhancing biofilm accumulation, and (3) modulate biofilm survival and persistence by inducing cytotoxicity that inhibit immune defenders. SE; staphylococcal enterotoxin.

**Figure 7 microorganisms-11-00089-f007:**
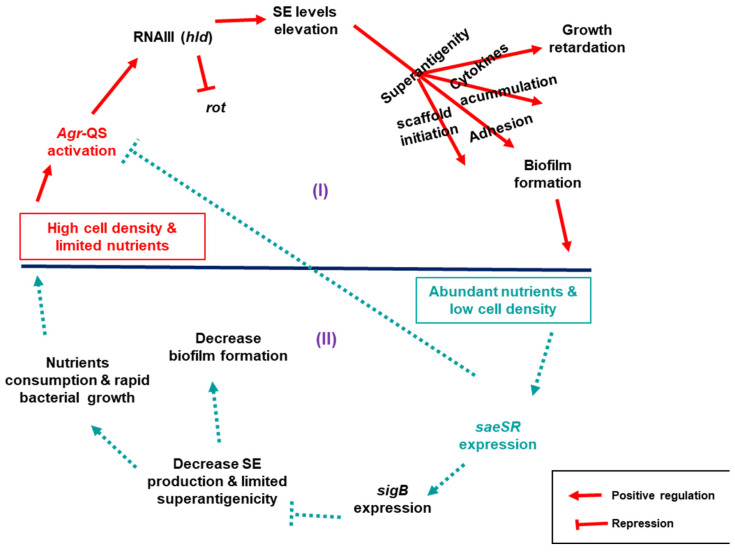
Proposed scenarios through which *saeSR*, *sigB*, and *Agr*-QS regulate SEs modulation of *S. aureus* growth and biofilm development in PCs. (I) High cell density & nutrient deficiency enviroment: Agr-QS is activated, signaling toxin expression by repressing *rot* through its effector RNAIII (*hld*), leading to growth retardation and biofilm matrix initiation, cell-to-cell adhesion, and biofilm development via superantigenic activity. (II) Nutrient abundance in low cell density environment: *saeSR* positively regulates the expression of *sigB*, which represses Agr-QS that consequentially decrease toxin production, enhancing rapid bacterial cells proliferation until the cell density increases again and Agr-QS is activated. Decrease in SE production reduces superantigenicity and, consequentially, biofilm accumulation.

**Table 1 microorganisms-11-00089-t001:** Biofilm description and development mechanisms: RNAseq-based differential expression (Log2 fold change) in PCs vs TSB of genes associated to different biofilm mechanisms in tested *S. aureus* isolates (CBS2016-05 = CBS; CI/BAC/25/13/W = CI/25; PS/BAC/169/17/W = PS/169; PS/BAC/317/16/W = PS/317).

Strains/Biofilm Parameters	CBS	CI/25	PS/169	PS/317	Gene
Biofilm Description	Strong	Weak	Strong	Weak	
PIA	−0.6	0.7	−1.1	0.7	*icaA*
	−0.3	0.7	−0.1	1.3	*icaB*
	−0.5	1.3	0.0	1.4	*icaC*
	−0.6	−3.0	0.1	0.3	*icaD*
Proteins/	3.9	1.2	0.8	3.3	*cidA*
eDNA	2.8	0.6	1.2	1.5	*lrgA*
	1.9	−0.2	−0.2	−0.2	*lrgB*
	0.8	0.5	1.1	0.4	*sasC*
Coa-fibrin	1.6	0.2	−0.6	−1.4	*clfA*
	2.7	2.1	−0.1	2.6	*clfB*
	1.1	1.1	0.0	0.9	*Coa*
	2.0	5.0	2.6	0.2	*sdrC*
	1.5	2.7	1.9	0.4	*sdrD*
Amyloid	−11.9	−6.1	−2.0	−8.0	*psma1*
	−8.0	−5.5	−1.2	−7.9	*psmb1*

**Table 2 microorganisms-11-00089-t002:** RNAseq-based differential expression (Log2 Fold change). Selected virulence factors (represented genes) in PCs against TSB of tested *S. aureus* isolates.

Strains/Virulence Factors	CBS	CI/25	PS/169	PS/317	Gene
Biofilm/	−0.5	0.0	−0.6	1.9	*alt*
Adhesin	0.7	−1.5	0.9	0.1	*can*
	−3.5	−6.0	0.7	0.2	*map*/*eap*
	0.7	−1.3	−0.4	−2.6	*ebp*
	0.2	−4.3	−1.8	0.2	*clfA*
Capsule	5.9	1.1	0.8	0.4	*capA*
	5.4	0.9	1.3	1.4	*capE*
	5.4	0.9	1.1	1.7	*capF*
	5.4	0.8	1.0	1.8	*capG*
Exotoxins	0.8	4.1	2.9	−1.9	*hla*
	0.5	0.2	0.1	0.1	*seg*
	2.3	NA	NA	NA	*she*
	5.8	−1.0	−1.0	2.6	*spa*
	6.7	3.0	4.3	1.3	*ssl14*
Global	−1.9	−3.5	0.1	−1.6	*agrA*
regulators	−7.4	−2.8	1.1	−2.2	*hld*
	1.0	1.0	0.1	0.6	*rot*
	0.1	−1.0	0.2	0.6	*sarA*
	−3.2	−5.7	0.2	−1.4	*saeR*
	−3.1	−5.8	0.2	−1.5	*saeS*
	1.1	1.1	−0.1	0.0	*slgB*

NA; not applicable (gene not encoded in the specific genome).

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
