# Peer review of "Staphylococcal Enterotoxins Enhance Biofilm Formation by Staphylococcus aureus in Platelet Concentrates"

_microorganisms, 2022, doi:10.3390/microorganisms11010089_

Round 1

Reviewer 1 Report

Chi and Ramirez-Arc have done a nice research work on investigated the novel relationship between SEs and growth, and SEs and biofilm formation by S. aureus in PCs using transcriptomics and molecular biology approaches. However, the manuscript needs few minor changes before accepting.

1.      Please add a visual diagram or flowchart of your methods and outcomes for better understanding to the reader

2.      What is the hypothesis of the study and clear primary and secondary objectives?

3.      What is the improved percentage of false-negativity with the current method?

4.      What are the limitations of the proposed current method?

5.      Please add future directives of the current method.

Author Response

Chi and Ramirez-Arc have done a nice research work on investigated the novel relationship between SEs and growth, and SEs and biofilm formation by S. aureus in PCs using transcriptomics and molecular biology approaches. However, the manuscript needs few minor changes before accepting.

Answer: We thank the Reviewer for the positive feedback

  1. Please add a visual diagram or flowchart of your methods and outcomes for better understanding to the reader

Answer: Figure 1 has been created to describe the experimental approach followed in the study. Remaining figures have been renumbered accordingly.

  1. What is the hypothesis of the study and clear primary and secondary objectives? Answer: We thank the reviewer for this question. The hypothesis and objectives of the study have been added at the end of the Introduction, on page 3 of the revised manuscript.

  1. What is the improved percentage of false-negativity with the current method?

Answer: As the study was aimed at advancing knowledge, the new findings have not been applied to improve detection of S. aureus in PCs, if this is the “current method” to which the Reviewer is referring to in this question. Further studies are undergoing by our group to investigate the use of enterotoxin detection to complement culture-based screening of PCs, but that study is out of the scope of this manuscript.

  1. What are the limitations of the proposed current method?

Answer: A paragraph addressing this has been added at the end of the Discussion of the revised manuscript (2nd last paragraph, lines 429 – 435). The study could be expanded by studying other enterotoxins and the addition of other S. aureus strains.

  1. Please add future directives of the current method.

Answer: A paragraph addressing this has been added at the end of the Discussion of the revised manuscript (last paragraph, lines 436 – 444). Future directions could to investigate the role that enterotoxins may play in platelet activation/cytokine production, with additional impact on biofilm formation. Also, detection of enterotoxins could complement current PC screening with culture methods.

Reviewer 2 Report

Dear Authors,

the presented study is fascinating and undoubtedly beneficial for the given field. The introduction is perfectly processed. The proposed findings are innovative. The experimental work is focused in a very unconventional way, and an even more exciting perspective is enabled.

 The text is written more or less precisely and without significant problematic points. I believe it delivers a clear and exciting message to the audience.

 I have only minor comments about the text:

1/ L65, 125, 128, 129, etc. - the inappropriate character used for degrees (°C)

2/ L87-98 - error in the font. The same applies to L131-155 and L159-172.

3/ There are no uniform graphs in the entire manuscript - the appearance of the graphs is different, and I ask for maximum unification (frame of graphs, etc.) - see Fig 1 - 4 and compare.

4/ L277 - incorrect entry in the text - end of line

5/ I recommend improving the conclusion text. 

Author Response

the presented study is fascinating and undoubtedly beneficial for the given field. The introduction is perfectly processed. The proposed findings are innovative. The experimental work is focused in a very unconventional way, and an even more exciting perspective is enabled.

The text is written more or less precisely and without significant problematic points. I believe it delivers a clear and exciting message to the audience.

Answer: We thank the Reviewer for the positive feedback.

 I have only minor comments about the text:

1/ L65, 125, 128, 129, etc. - the inappropriate character used for degrees (°C)

Answer: The degree symbol has been corrected throughout the revised manuscript.

2/ L87-98 - error in the font. The same applies to L131-155 and L159-172.

Answer: Font type has been unified throughout the revised manuscript.

3/ There are no uniform graphs in the entire manuscript - the appearance of the graphs is different, and I ask for maximum unification (frame of graphs, etc.) - see Fig 1 - 4 and compare.

Answer: Graphs have been unified; figures 1-5 are unframed in the revised manuscript.

 4/ L277 - incorrect entry in the text - end of line

Answer: Text corrected in L294 and L298 of the revised manuscript.

5/ I recommend improving the conclusion text. 

Answer: We appreciate the Reviewer’s recommendation, and the Conclusion has been improved in the revised manuscript.
